# On the limits of cross-domain generalization in automated X-ray prediction

**Joseph Paul Cohen**                 joseph@josephpcohen.com
**Mohammad Hashir**                 mkhan31@vols.utk.edu
**Rupert Brooks**                 rupert.brooks@nuance.com
**Hadrien Bertrand**                hadrien.bertrand@mila.quebec
*Mila, Université de Montréal*

## Abstract

This large scale study focuses on quantifying what X-rays diagnostic prediction tasks generalize well across multiple different datasets. We present evidence that the issue of generalization is not due to a shift in the images but instead a shift in the labels. We study the cross-domain performance, agreement between models, and model representations. We find interesting discrepancies between performance and agreement where models which both achieve good performance disagree in their predictions as well as models which agree yet achieve poor performance. We also test for concept similarity by regularizing a network to group tasks across multiple datasets together and observe variation across the tasks. All code is made available online and data is publicly available:
https://github.com/mlmed/torchxrayvision

**Keywords:** X-rays diagnostic, deep learning, generalization

## 1. Introduction

This work studies the generalization performance of current chest X-rays prediction models when trained and tested on X-rays image datasets from different institutions that were annotated by different clinicians or labelling tools. By doing so, we aim to provide supporting evidence for which tasks are reliable/consistent across multiple different datasets. Indeed, it seems there are limits to the performance of systems designed to replicate humans which is consistent with the evidence that human radiologists often don't agree with each other. Recent research has discussed generalisation issues (Pooch et al., 2019; Yao et al., 2019; Baltruschat et al., 2019) however it is not clear exactly what the cause of the problem is. We enumerate some possibilities:

- Errors in labelling as discussed by Oakden-Rayner (2020) and Majkowska et al. (2019), in part due to automatic labellers.
- Discrepancy between the radiologist's vs clinician's vs automatic labeller's understanding of a radiology report (Brady et al., 2012).
- Bias in clinical practice between doctors (Busby et al., 2018) or limitations in objectivity (Cockshott and Park, 1983; Garland, 1949).
- Interobserver variability (Moncada et al., 2011). It can be related to the medical culture, language, textbooks, or politics. Possibly even conceptually (e.g. footballs between USA and the world ⚽🏈).

Formally we have pairs of X-ray images, $x_i$, and corresponding task labels, $y_i$, drawn from some joint distribution $p(x, y)$ for a given population. Our learning methods estimate $p(y|x)$, but may not generalize well when the joint distribution changes due to, for example, different X-ray machines or variable patient characteristics between different populations. There are several different cases that can give rise to variations in $p(x, y)$ and we will use the terminology of (Moreno-Torres et al., 2012) to describe them. Approaches for generalizing medical image models (e.g. (Pooch et al., 2019)) have assumed $p(y|x)$ to be constant and concentrated on *covariate shift* (where $p(x)$ varies) and *prior probability shift* (where $p(y)$ varies). We present evidence that $p(y|x)$ is not consistent and what is considered the "ground truth" is subjective; *concept shift* in the terminology of (Moreno-Torres et al., 2012). This forces us to consider $p(y|x, c)$ where $c$ conditions the prediction. Our experiments suggest that this conditioning is not only related to bias from the population but is due to other factors. This presents a new challenge to overcome when developing diagnostic systems as, under the current formulation, it may be impossible to train a system that will generalize.

To address this issue Majkowska et al. (2019) relabeled a subset of the NIH dataset images for 4 labels using 3 raters. On these images their raters didn't agree with each other for "Airspace opacity" 10% of the time and "Nodule/mass" 6% of the time[1]. When looking at NIH images which have been used in other datasets and relabelled for the same pathologies (Appendix Figure A.1) we find generally poor agreement with NIH labels for positive predictions and F1 scores as low as 10% (for Pneumonia). The Kaggle and Google relabelings show better, but very far from perfect, agreement on the one category where they overlap (Opacity, F1: 73%).

When creating the MIMIC-CXR dataset, Johnson et al. (2019) used two different automatic label extraction methods. Between these methods the most disagreement was 0.6% for "Fracture" (when only considering positive and negative labels) or 2.6% for Cardiomegaly (when including uncertain and no prediction as well). They also evaluated a subset of the radiology reports with a board certified radiologists which found that a lowest agreement of 0.462 F1 for "Enlarged Cardiomediastinum" which can possibly be explained by uncertainty about what cardio-thoracic ratio (CTR) is clinically relevant (Zaman et al., 2007).

These studies indicate that automatic labelling tools are consistent with each other and the issue likely is related to the well known problem of interobserver variability. In order to mitigate this problem we focus on studying its impact on the current Deep Learning approaches.

**Our approach:** In this work we analyze models trained on four of the largest public datasets utilizing over 200k unique chest X-rays after filtering for one AP or PA view per patient. A study like this is needed as these systems are being built and evaluated now (Cohen et al., 2019; Qin et al., 2019; Baltruschat et al., 2019; Hwang et al., 2019; Rubin et al., 2018; Yao et al., 2019; Putha et al., 2018). This work is further motivated by the use of these models in populations much different than their training population such as in (Qin et al., 2019) where systems such as qXR (developed in India) is applied to images from Nepal and Cameroon.

There are many issues that could prevent a model from generalizing. For example: overfitting to artifacts of the training data (Zech et al., 2018), concepts can vary between

---

1. We calculate these statistics from the published file individual_readers.csv. If there was not unanimous agreement between the 3 raters this is considered disagreement.

training labels and external data, training data may not be a representative sample of external data, and the models could be learning very superficial image statistics (Jo and Bengio, 2017).

The paper is structured into three sections: performance, agreement, and representation. The performance section §4 studies performance of models trained on one dataset and evaluated on others. The agreement section §5 studies how much predictions from models trained on one dataset agree with the predictions of other models trained using other datasets for the same task. Finally a representation section §6 studies how well the representations in the neural networks differ between the models. All code is made available online[2] and data is publicly available.

## 2. Data

We use the following datasets: **NIH** aka Chest X-ray14 (Wang et al., 2017), **PC** aka PadChest (Bustos et al., 2019), **CheX** aka CheXpert (Irvin et al., 2019), **MIMIC-CXR** (Johnson et al., 2019), **OpenI** (Demner-Fushman et al., 2016), **Google** (Majkowska et al., 2019), **Kaggle** aka the RSNA Pneumonia Detection Challenge[3]. Full details of the data are located in Appendix §A. 18 common labels were identified by manually reviewing the descriptions of the provided labels in each dataset. Code is provided which details the exact mapping online. We release a framework to load these datasets in a canonical way for further experimentation called torchxrayvision (Cohen et al., 2020). To align the datasets we resize the images to $224 \times 224$ pixels as is standard for methods on these datasets Rajpurkar et al. (2017). We did not want to confuse the issue by changing the architecture and strategy from previous work. The images are also center cropped if the aspect ratio is uneven (as to not stretch the images) and the pixel values are scaled between $[-1024, 1024]$ so that bit depth of the images is uniform.

## 3. Models

DenseNets (Huang et al., 2017) have been shown to be the best architecture for X-rays predictive models (Rajpurkar et al., 2017). Training was standard with other similar work. To take into account that only some labels are present with the recent 2019+ datasets the loss is computed only for the available labels and other outputs are ignored. An ensemble of three models are trained for each dataset and the results averaged to reduce noise.

Due to label imbalance the performance for tasks which are overrepresented receive less focus by the loss function. In order to alleviate this the weight for each task is balanced based on the frequency of that task in the dataset. Each task $t$ is given a weight $w_t$ based on the following formula where $c_t$ is the count of samples with positive samples for task $t$ and $\bar{c}$ is the average count. The intuition here is that $\max_i(c_i) - c_t$ will be 0 for at least one task so $\bar{c}$ pushes up the minimum weight while $\frac{\alpha_t}{\max_i(\alpha_i)}$ normalizes this value to be between 0 and 1.

$$w_t = \frac{\alpha_t}{\max_i(\alpha_i)}, \qquad \alpha_t = \max_i(c_i) - c_t + \bar{c} \tag{1}$$

---

2. https://github.com/mlmed/torchxrayvision
3. https://www.kaggle.com/c/rsna-pneumonia-detection-challenge

In order to calibrate the output of the model so that they can be compared a piecewise linear transformation Eq. 2 is applied. The transformation is chosen so that the best operating point corresponds to 50%. For each disease, we computed the optimal operating point by maximizing the difference (*True positive rate - False positive rate*). It corresponds to the threshold which maximizes the informedness of the classifier (Powers, 2011). This is computed with respect to the test set being evaluated so the model is the most optimal it can be. With this we remove miscalibration as a reason for generalization error.

$$f_{opt}(x) = \begin{cases} \frac{x}{2opt} & x \leq opt \\ 1 - \frac{1-x}{2(1-opt)} & otherwise \end{cases} \tag{2}$$

It is important to note that Eq. 2 requires an operating point which is not the same across all datasets. For example if we calibrate on NIH data so a prediction of 0.5 is the optimal decision boundary (FPR TPR tradeoff) for that dataset and then apply the model to PADCHEST the optimal decision boundary will be different and possibly 0.8. This means a prediction of 0.79 should be considered negative, and the model should be calibrated using Eq. 2 so that $0.8 \rightarrow 0.5$. In this study we calibrate each model based on the test data it is evaluating at that moment in order to remove this issue from consideration and assume the model is operating with optimal calibration. Each of the models in the ensemble is calibrated separately, and their calibrated output is averaged.

Data augmentation was used to improve generalization. According to best results in Cohen et al. (2019) (and replicated by us) each image was rotated up to 45 degrees, translated up to 15% and scaled larger of smaller up to 10%.

## 4. Performance

The most basic analysis to evaluate how well models generalize is to look at the performance outside their training data. In Figure 1 a model is trained on each dataset's training subset and then evaluated on the other dataset's testing subsets. AUC is used to determine the performance per task as it accounts for imbalance in labels. Many combinations are not possible because the datasets do not overlap completely and we aim to include as many labels as possible.

The experiments show the best generalization for the tasks Cardiomegaly, Edema, and Effusion. It also seems there is reasonable generalization for Atelectasis, Consolidation, Emphysema, Hernia, and Lung Opacity. The worst generalization performance can be seen for Infiltration where it is inverted between the PC and NIH datasets. Pneumonia indicates that the NIH model performs poorly and other models also perform poorly on the NIH while performing well on other datasets. For Fracture all models applied to the hand labelled NIH_Google dataset perform very poorly while much better on their own test set than others. Pneumothorax also indicates better performance on a models test set than others but does perform well on the hand labelled NIH_Google dataset.

In Figure 1 the "All" model which is trained on all datasets combined outperforms almost all other models (with the exception of Pneumonia on NIH). However, this result is not due to better generalization but is due to the inclusion of the training data which comes from the same domain. To verify this, similar to Yao et al. (2019), the performance on a test set is evaluated in Figure 2 when leaving the test set domain out of training.

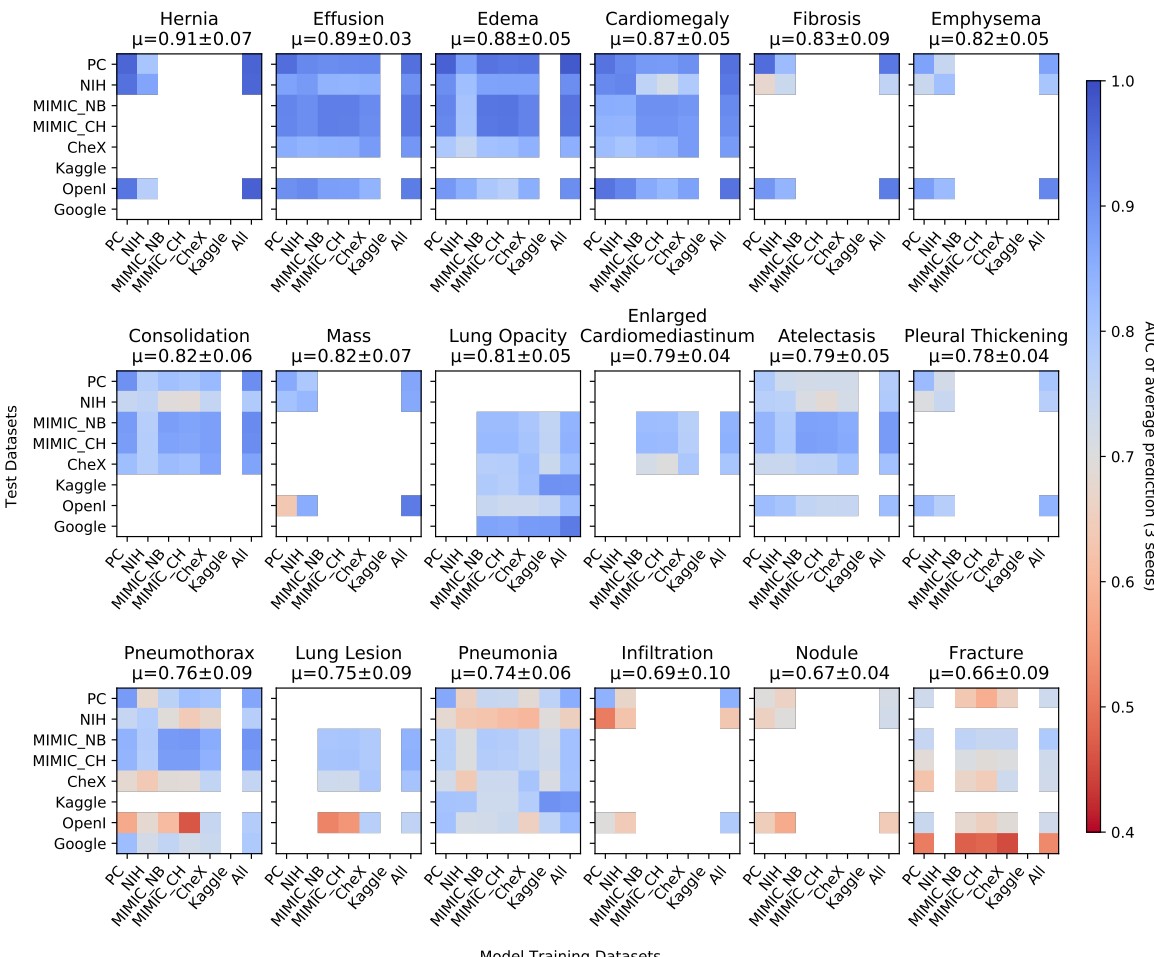

Figure 1: AUC of each model on each dataset. All valid combinations of model and dataset are computed where a model was trained on the specific label and that label exists in the target dataset. A white cell means it cannot be computed due to missing labels in train or test dataset. The outputs of 3 models are averaged together to reduce noise. Each of the 3 models is trained on the same data with different weight initialization.

## 5. Agreement

We use the Cohen's Kappa score (Cohen, 1960) in order to calculate the agreement between raters, or in our case networks trained on a specific dataset. $\kappa = \frac{p_o - p_e}{1 - p_e}$. A Kappa of 0 indicates only chance agreement and 1 indicates total agreement. A Kappa of 40% is considered moderate while 70% is considered excellent (Moncada et al., 2011).

In Figure 3 agreement is poor for labels which are not common. Agreement between the models themselves when trained with different seeds is between 75% and 86% Kappa (in appendix Figure A.2) indicating that is the upper bound. The tasks are ordered from left to right based on their generalization performance as evaluated in §4. An unexpected finding

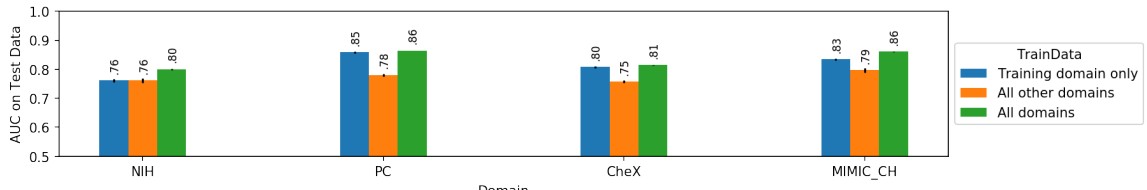

Figure 2: A leave one domain out evaluation. Here blue represents a model trained on only the training data of the domain under test. The orange bar represents a model trained on all domains except the domain under test. Finally the green bar represents a model trained on all domains including the domain under test. The average AUC over all tasks are shown. Three seeds are used to initialize the models and the mean and stdev is shown.

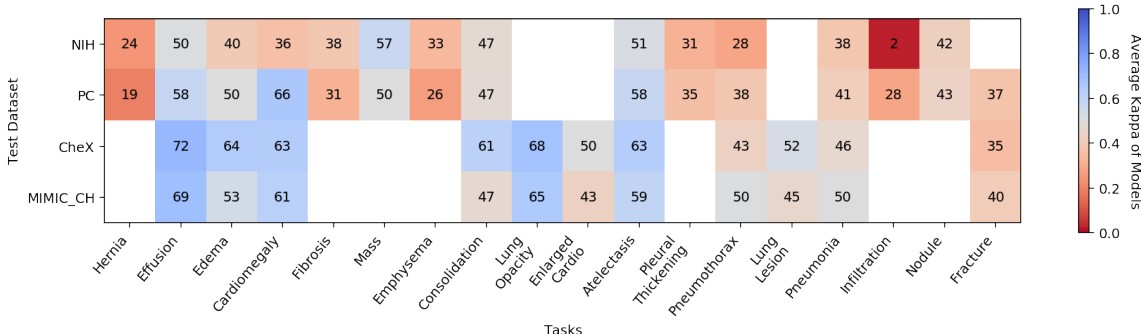

Figure 3: Kappa inter-rater variability for pairwise comparisons given each testset. A white cell means it cannot be computed due to missing labels in train or test dataset. An ensemble of 3 models is used to reduce noise. For each task the Kappa score is calculated between the model trained on that data and other models which are trained to predict that task.

is that Cardiomegaly is generally in agreement except by the NIH model which seems to perform well except for the MIMIC_CH dataset. These results are concerning as models can disagree yet still perform well.

Some tasks can disagree yet achieve high AUC which others have strong agreement yet have low AUC. The outputs between two such models and tasks are studied in Figure 4.

## 6. Representation

We can also look at how the representation in a model changes between training datasets. In this experiment we train a network which has an output that represents a single dataset-task combination resulting in a weight vector for each (5 datasets × 18 tasks = 90 outputs). With this approach each image is processed into a feature vector of dimension 1024. A classifier layer is applied to these feature vectors, where each task output is determined by a sigmoid (logistic function). These vectors are updated in order to improve a single task and

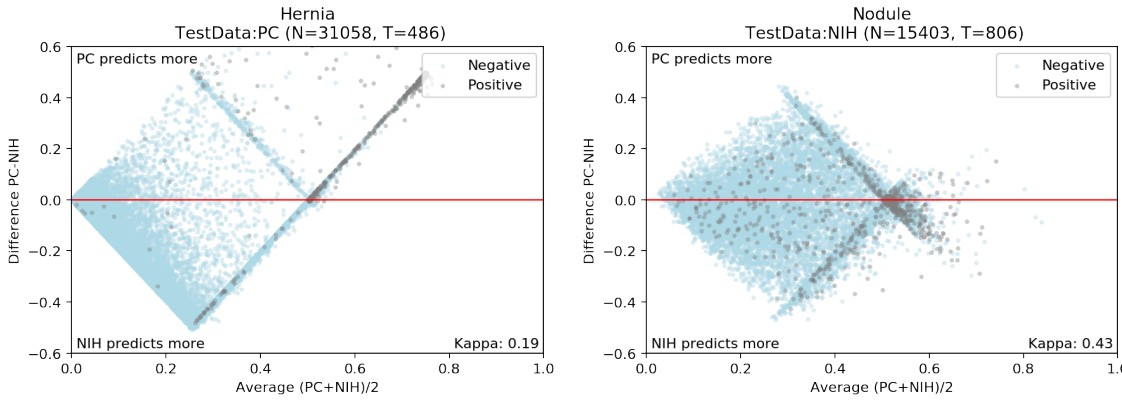

($a$) Good performance but poor agreement    ($b$) Bad performance but high agreement

Figure 4: Bland Altman plots showing agreement of the model outputs. The red line indicates where optimal agreement should be. The model outputs are calibrated so that 0.5 is the operating point of the AUC and therefore is the optimal threshold. This calibration causes the diamond artifacts when plotted.

are therefore independent of each other (ignoring transfer learning via multitask training). During training each vector is updated only with respect to their datasets. If these weight vectors are the same between two tasks then their predictions will be identical. Because the logistic function is a relatively linear transformation the distance between these vectors is meaningful and can explain similarity between tasks.

In Figure 5 the first 2 components of the Principal component analysis (PCA) (Pearson, 1901) are plotted for every domain-task vector. This is a linear dimensionality reduction method so distances are real unlike a t-SNE (Maaten, 2009). We can observe some similarity of the tasks such as Cardiomegaly and Effusion but generally the vectors are very different.

We can add an L2 regularizer that encourages the weight vectors of the same task to be close to each other. This is added to the objective function so the model is simultaneously learning to make predictions while it is trying to align these weight vectors. In the lower figure of Figure 5 the results of training with this regularization are shown. We can see that even with this pressure to align weight vectors, some tasks do not merge into a single vector as Mass, Nodule, Fibrosis, Lung Lesion, and Pleural Thickening.

The more variation between these task vectors the more evidence that for the same feature vector a different prediction must be made. This implies that the differences between the datasets during training have caused the network to diverge in its representation of a task and produce different results. These differences are viewed explicitly in Figure 6 where the differences between weight vectors have been averaged over 3 seeds and normalized relative to the other tasks.

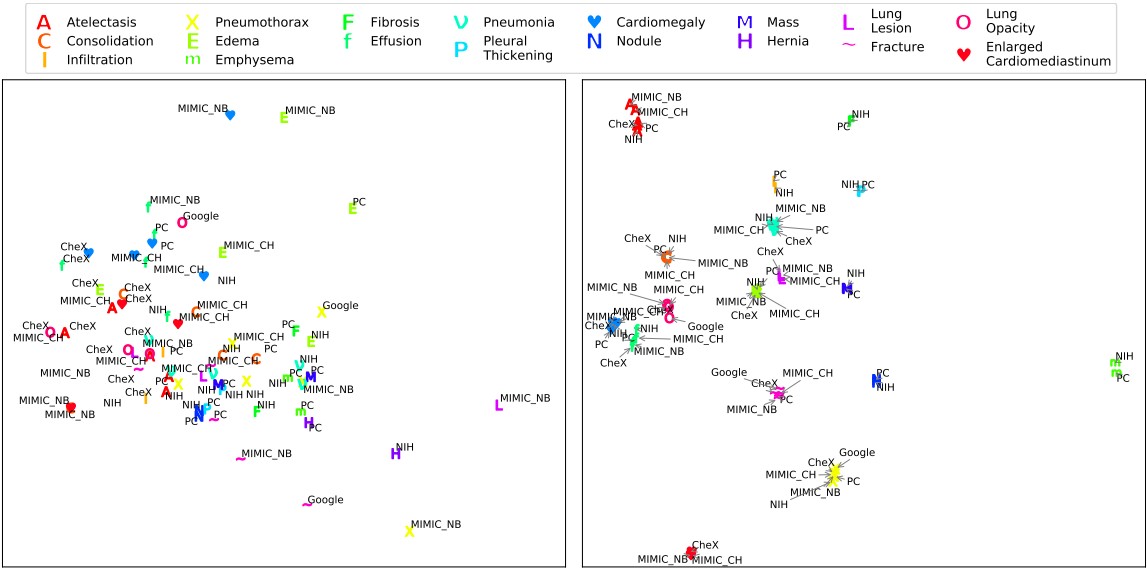

Figure 5: Two models trained so that each output represents a single dataset-task combination resulting in a weight vector for each. A PCA of these weight vectors is shown. The left shows the normal training case while the right shows the result when trained with regularization so that all vectors for the same task are similar (L2 distance).

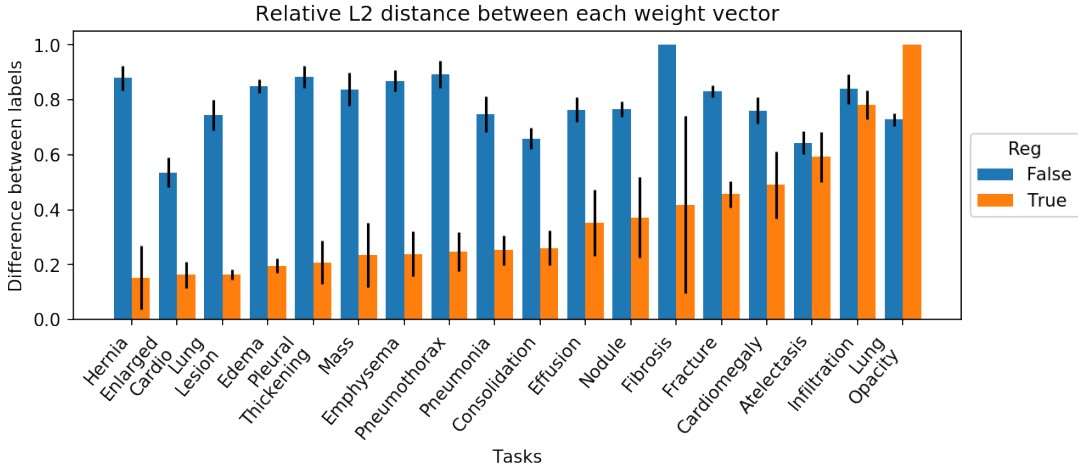

Figure 6: Distances between regularized and non-regularized models. Tasks are sorted by their distance while regularized. The average and standard deviation of 3 models trained with different seeds and data splits is shown.

## 7. Discussion

This work presents evidence that the community may want to focus on concept shift over covariate shift in order to improve generalization of chest X-ray prediction models. If covariate shift was only present then it is unexpected that we would observe over half of the tasks perform well while the remaining have very variable results. Our results, specifically the discrepancy between model prediction agreement and performance, raise more questions that warrant further study.

In order to address this problem it seems that better automatic labeling may not be the solution as the bias is likely at the level of different schools of thought, general disagreement between radiologists, and subjectivity in what is clinically relevant to include in a report.

If these networks are anything like doctors then discrepancy, difference of opinion, and errors are unavoidable (Siegle et al., 1998; Brady et al., 2012; Brady, 2017; Soffa et al., 2004). As these models are only trained to capture the conditional distribution defined by the training distributions they will carry with them the bias of the data. When building these into tools which influence clinical outcomes we shouldn't accept that model predictions reflect our own idea of a medical concept. We should consider each task prediction as defined by its training data such as "NIH Pneumonia". One can present the output of multiple models to a user with information about the specific context and origin of that model.

We assert that a solution is not to train on a local data from a hospital that the tool will be deployed in. We have shown that even though a model trained using all datasets performs well it does not reflect true generalization performance. It follows that we should not be fine-tuning models on local distributions as it is likely only adapting to the local biases in the data which may not match the reality in the images.

## 8. Limitations

Only labels associated with each dataset are used and the outcomes of the patients are not considered. This would be relevant for establishing the risk of disagreement for specific tasks. We only use the AP/PA views and ignore the lateral views which many contain needed features of a finding as discussed by Bertrand et al. (2019) and Hashir et al. (2020).

## Acknowledgments

We thank Mathieu Germain, Chin-Wei Huang, Karsten Roth, Joseph Viviano, Lan Dao, Ronald Summers, Yoshua Bengio, Gabriele Prato, and Michaël Chassé for their feedback. We also specifically thank Chin-Wei Huang for help selecting perfect icons for Figure 5. This work utilized the supercomputing facilities managed by Compute Canada and Calcul Quebec. We thank AcademicTorrents.com for making data available for our research.

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

# Appendix A. Data details

- **NIH** (30k images) From the Clinical Center, Bethesda, Maryland, USA. The Chest-X-rays14 dataset released by the NIH (Wang et al., 2017). It was automatically labelled using the NegBio labeller.

- **PC** (62k images) Aka PadChest, from Hospital San Juan de Alicante, Alicante, Spain (Bustos et al., 2019). The images have been labeled with various types of radiological findings and differential diagnoses, with 27% of the annotations created manually by physicians and the rest extracted from the report by an RNN. Since the PadChest dataset defines a hierarchy of labels, we mapped the labels to their respective top level.

- **CheX** (64k images) Aka CheXpert, from the Stanford Hospital, Palo Alto, CA, USA (Irvin et al., 2019). This dataset introduces a custom labeller called the "CheXpert labeler".

- **MIMIC-CXR** (45k images) From the Beth Israel Deaconess Medical Center in Boston, MA, USA (Johnson et al., 2019). Labels were extracted and are provided from two automatic labellers, both the CheXpert and the NIH NegBio labeller. MIMIC_CH refers to the CheXpert labeller and MIMIC_NB refers to the NIH NegBio labeller.

- **OpenI** (3267 images) From the Indiana University hospital network (Demner-Fushman et al., 2016). The MeSH automatic labeller was used.

- **Google** (1695 images) Images from the NIH data were relabeled manually (Majkowska et al., 2019) for 4 labels. We don't use the "mass/nodule" label as it does not align with our standardization of labels.

- **Kaggle** (30227 images) From the Kaggle Pneumonia Detection Challenge[4]. Each image was hand labelled by a single radiologist for the presence of lung opacity. This label is included as both Lung Opacity and Pneumonia.

Detailed dataset information is in Appendix Table A.1.

---

4. https://www.kaggle.com/c/rsna-pneumonia-detection-challenge

| Dataset | NIH | PC | CheX | Google | MIMIC_CH | MIMIC_NB | OpenI | Kaggle |
|---|---|---|---|---|---|---|---|---|
| Atelectasis | 1702/29103 | 2441/59674 | 12691/14317 | - | 4077/30954 | 4048/32058 | 271/2996 | - |
| Cardiomegaly | 767/30038 | 5390/56725 | 9099/17765 | - | 3743/32312 | 3275/33431 | 185/3082 | - |
| Consolidation | 427/30378 | 494/61621 | 5390/22504 | - | 816/32297 | 762/33564 | - | - |
| Edema | 82/30723 | 108/62007 | 14929/20615 | - | 1157/33610 | 1121/34731 | 50/3217 | - |
| Effusion | 1280/29525 | 1637/60478 | 20640/23500 | - | 3713/33401 | 3595/34489 | 120/3147 | - |
| Emphysema | 265/30540 | 546/61569 | - | - | - | - | 84/3183 | - |
| Enlarged Cardio | - | - | 5181/20506 | - | 692/31505 | 660/32641 | - | - |
| Fibrosis | 571/30234 | 341/61774 | - | - | - | - | 17/3250 | - |
| Fracture | - | 1665/60450 | 4250/14948 | 60/1635 | 972/30961 | 696/32320 | 78/3189 | - |
| Hernia | 83/30722 | 988/61127 | - | - | - | - | 41/3226 | - |
| Infiltration | 3604/27201 | 4438/57677 | - | - | - | - | 66/3201 | - |
| Lung Lesion | - | - | 4217/14422 | - | 1321/31033 | 1271/32187 | 3/3264 | - |
| Lung Opacity | - | - | 30873/15675 | 601/1094 | 5426/31175 | 5301/32371 | 327/2940 | 9555/20672 |
| Mass | 1280/29525 | 507/61608 | - | - | - | - | 6/3261 | - |
| Nodule | 1661/29144 | 2194/59921 | - | - | - | - | 68/3199 | - |
| Pleural_Thickening | 763/30042 | 2076/60039 | - | - | - | - | 30/3237 | - |
| Pneumonia | 168/30637 | 2051/60064 | 2822/14793 | - | 2176/33347 | 2042/34479 | 68/3199 | 9555/20672 |
| Pneumothorax | 269/30536 | 98/62017 | 4311/32685 | 72/1623 | 560/33651 | 500/34760 | 14/3253 | - |

Table A.1: Counts of samples in datasets. What is shown is positive/negative. some datasets omit labels while others have a negative value for each dataset.

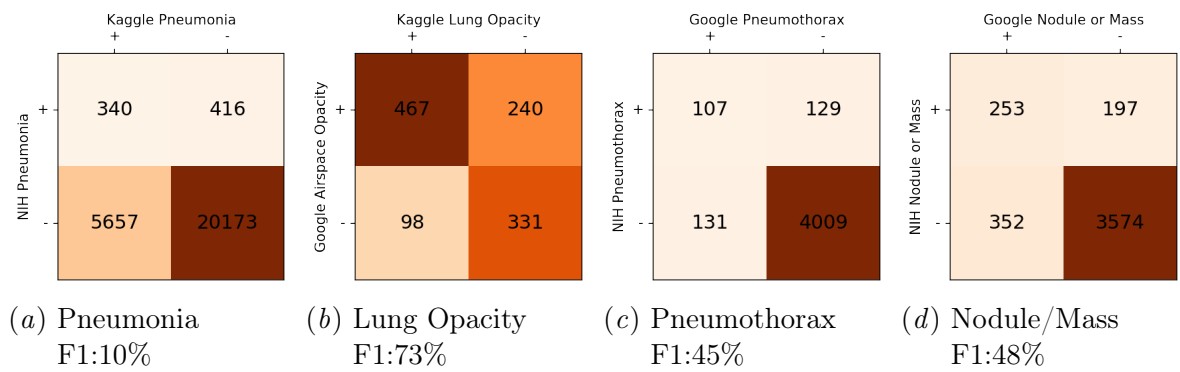

(*a*) Pneumonia
F1:10%

(*b*) Lung Opacity
F1:73%

(*c*) Pneumothorax
F1:45%

(*d*) Nodule/Mass
F1:48%

Figure A.1: Label agreement between different datasets which use NIH images. Samples from the NIH dataset were relabelled in the Kaggle and Google datasets. The Google dataset explicitly lists the corresponding NIH image, while the Kaggle dataset could be rematched based on pixel similarity. This figure shows the confusion matrices for images which were labelled by two of the datasets.

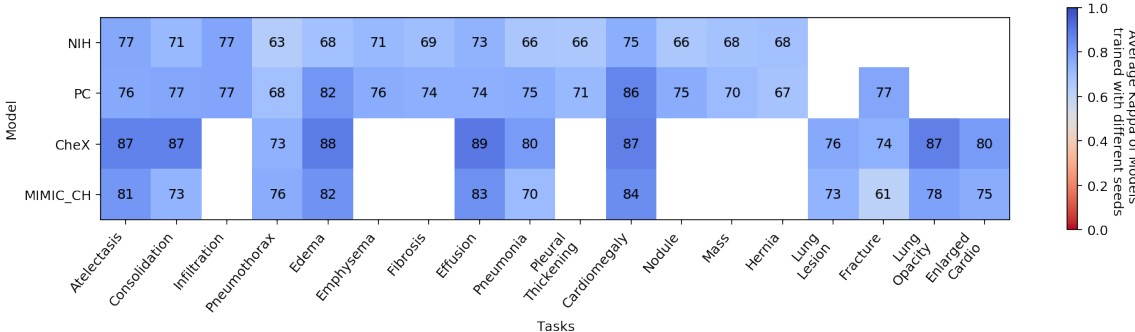

Figure A.2: Kappa Inter-rater variability for pairwise comparisons given each model over the 3 seeds.

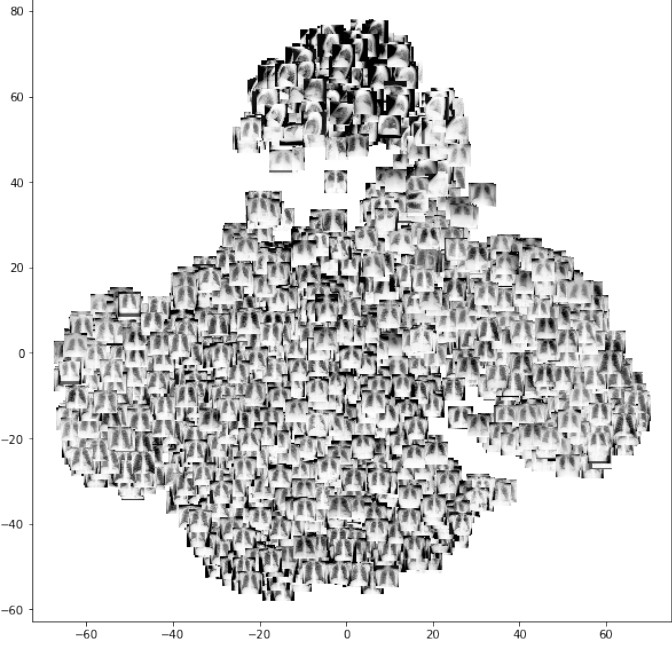

Figure A.3: t-SNE of features extracted from OpenI images in order to determine PA view from lateral view images.

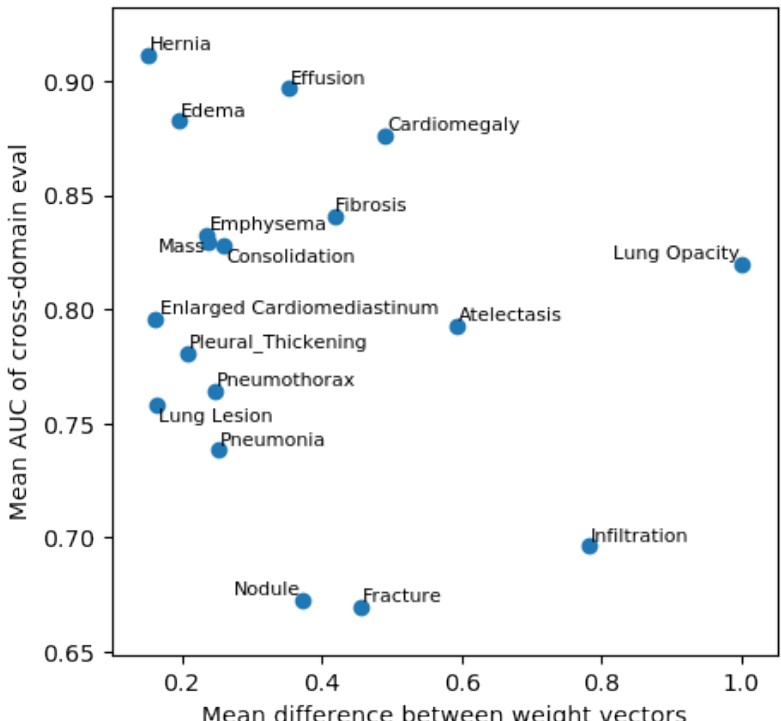

Figure A.4: Relationship between generalization performance and similarity between weight vectors.

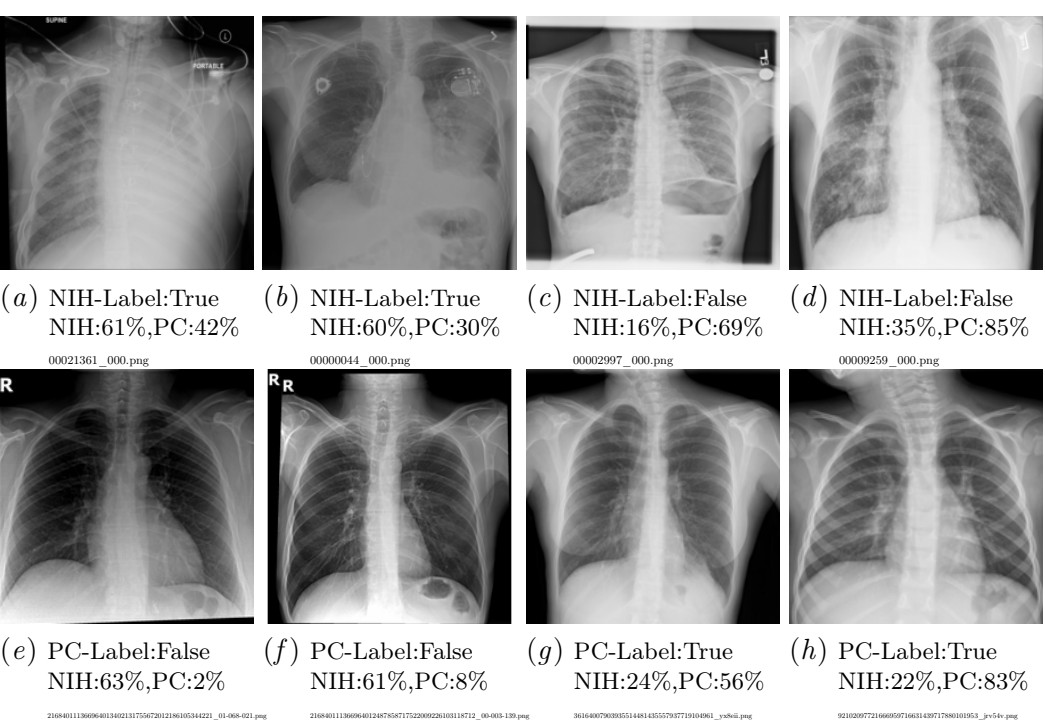

$(a)$ NIH-Label:True
NIH:61%,PC:42%

$(b)$ NIH-Label:True
NIH:60%,PC:30%

$(c)$ NIH-Label:False
NIH:16%,PC:69%

$(d)$ NIH-Label:False
NIH:35%,PC:85%

$(e)$ PC-Label:False
NIH:63%,PC:2%

$(f)$ PC-Label:False
NIH:61%,PC:8%

$(g)$ PC-Label:True
NIH:24%,PC:56%

$(h)$ PC-Label:True
NIH:22%,PC:83%

Figure A.5: Images most in disagreement for label Infiltration. Left: NIH model predicts higher, Right: PC model predicts higher. Top row is NIH dataset images and bottom row is from PC. All images are labelled as Infiltration for their respective dataset. The probability of each model is shown below the image. The outputs are calibrated so 50% is the operating point for each model.

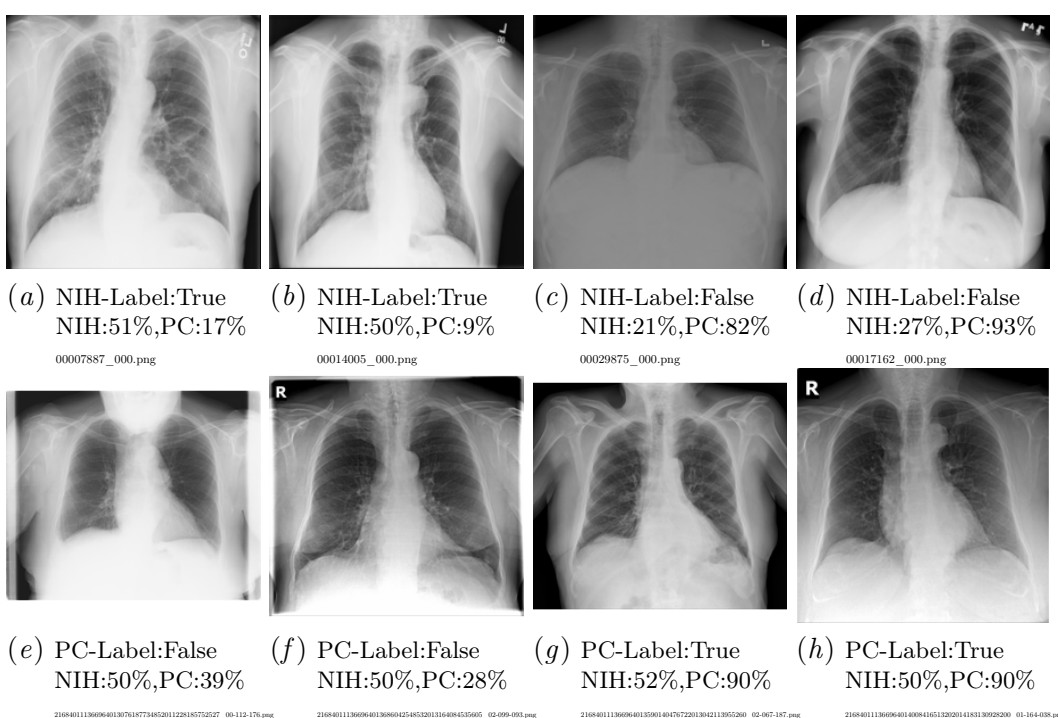

($a$) NIH-Label:True NIH:51%,PC:17%
00007887_000.png

($b$) NIH-Label:True NIH:50%,PC:9%
00014005_000.png

($c$) NIH-Label:False NIH:21%,PC:82%
00029875_000.png

($d$) NIH-Label:False NIH:27%,PC:93%
00017162_000.png

($e$) PC-Label:False NIH:50%,PC:39%
216840111366964013076187734852011228185752527_00-112-176.png

($f$) PC-Label:False NIH:50%,PC:28%
216840111366964013686042548532013164084535605_02-099-093.png

($g$) PC-Label:True NIH:52%,PC:90%
216840111366964013590140476722013042113955260_02-067-187.png

($h$) PC-Label:True NIH:50%,PC:90%
216840111366964014008416513202014183130928200_01-164-038.png

Figure A.6: Images most in disagreement for Hernia in the PC and NIH datasets. Left: NIH model predicts higher, Right: PC model predicts higher. The probability of each model is shown below the image. The outputs are calibrated so 50% is the operating point for each model.

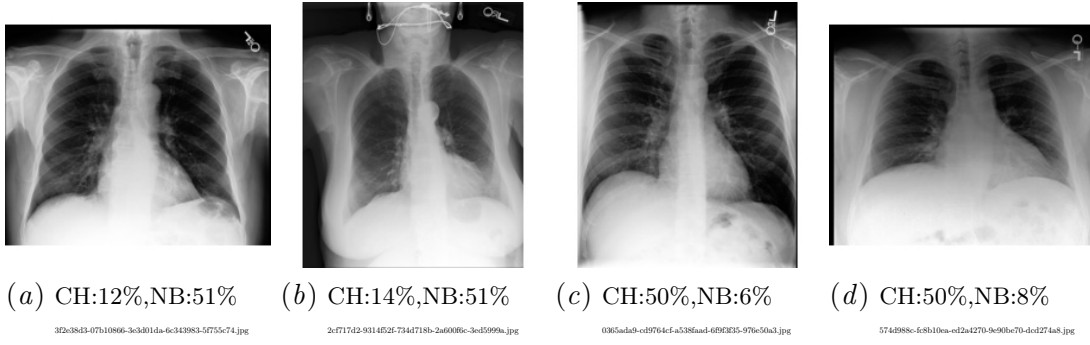

(*a*) CH:12%,NB:51%  (*b*) CH:14%,NB:51%  (*c*) CH:50%,NB:6%  (*d*) CH:50%,NB:8%

3f2c38d3-07b10866-3e3d01da-6c343983-5f755c74.jpg 2cf717d2-9314f52f-734d718b-2a600f6c-3ed5999a.jpg 0365ada9-cd9764cf-a538faad-6f9f3f35-976e50a3.jpg 574d988c-fc8b10ea-ed2a4270-9e90be70-dcd274a8.jpg

Figure A.7: Images most in disagreement for Fracture in the MIMIC-CXR dataset. All images have the ground truth labelling CH-Label:True NB-Label:False. Left: NB model predicts higher, Right: CH model predicts higher. The probability of each model is shown below the image. CH indicates the probability output by the model trained using the CheXpert labels and NB indicates the probability output by the model trained using the NegBio labels. The outputs are calibrated so 50% is the operating point for each model. There was only one sample where the NB label was true and CH label was false and it is not shown as the networks both strongly predicted a negative score.

